# Synergistic Mechanisms of Constituents in Herbal Extracts during Intestinal Absorption: Focus on Natural Occurring Nanoparticles

**DOI:** 10.3390/pharmaceutics12020128

**Published:** 2020-02-03

**Authors:** Qing Zhao, Xin Luan, Min Zheng, Xin-Hui Tian, Jing Zhao, Wei-Dong Zhang, Bing-Liang Ma

**Affiliations:** 1Department of Pharmacology, School of Pharmacy, Shanghai University of Traditional Chinese Medicine, Shanghai 201203, China; 15221811720@163.com (Q.Z.); zhengminpkyaodong@163.com (M.Z.); breadsweet@126.com (J.Z.); 2Institute of Interdisciplinary Integrative Medicine Research, Shanghai University of Traditional Chinese Medicine, Shanghai 201203, China; luanxin@shutcm.edu.cn (X.L.); tianxinhui@126.com (X.-H.T.); 3School of Pharmacy, Second Military Medical University, Shanghai 200433, China

**Keywords:** herbal extract, pharmacokinetic synergy, intestinal absorption, secondary metabolites, nanoparticle, natural deep eutectic solvent

## Abstract

The systematic separation strategy has long and widely been applied in the research and development of herbal medicines. However, the pharmacological effects of many bioactive constituents are much weaker than those of the corresponding herbal extracts. Thus, there is a consensus that purer herbal extracts are sometimes less effective. Pharmacological loss of purified constituents is closely associated with their significantly reduced intestinal absorption after oral administration. In this review, pharmacokinetic synergies among constituents in herbal extracts during intestinal absorption were systematically summarized to broaden the general understanding of the pharmaceutical nature of herbal medicines. Briefly, some coexisting constituents including plant-produced primary and secondary metabolites, promote the intestinal absorption of active constituents by improving solubility, inhibiting first-pass elimination mediated by drug-metabolizing enzymes or drug transporters, increasing the membrane permeability of enterocytes, and reversibly opening the paracellular tight junction between enterocytes. Moreover, some coexisting constituents change the forms of bioactive constituents via mechanisms including the formation of natural nanoparticles. This review will focus on explaining this new synergistic mechanism. Thus, herbal extracts can be considered mixtures of bioactive compounds and pharmacokinetic synergists. This review may provide ideas and strategies for further research and development of herbal medicines.

## 1. Introduction

Herbal medicines such as those used in traditional Chinese medicines (TCMs) are still used globally, especially to meet the healthcare needs in developing countries [1]. Herbal medicines are commonly used as a crude extract, either dried or not, which is essentially a mixture of both the primary and secondary metabolites of the original plants. Primary metabolites (PMs) include plant-produced proteins, lipids, amino acids, and sugars [2]. Secondary metabolites (SMs) refer to plant-produced small molecular compounds, bioactive or otherwise [3]. Due to their intrinsic complexity, the quality control of herbal medicines is very challenging [4]. Therefore, active constituents rather than crude extracts are now preferred in the research and development of new drugs. Direct phytochemical isolation or bioactivity-guided fractionation [5] of herbal extracts has long and widely been conducted to identify and enrich active constituents. Fortunately, herbal medicines are rich sources of compounds with novel structures. From 1981 to 2014, approximately one-third of newly approved small-molecule drugs are natural or naturally derived products [6].

However, after separation and purification from herbal extracts, the pharmacological effects of many bioactive constituents diminish or even disappear [3]. For example, despite the clinically verified pharmacological effects of artemisinin and its worldwide application for malaria treatment in humans, dried whole-plant *Artemisia annua* L. is superior to artemisinin in slowing the evolution of malaria drug resistance and overcoming resistance [7]. The decrease of pharmacological effects is closely related to the loss of pharmacokinetic synergies among constituents after the herbal extract is purified [8]. Some bioactive constituents exhibit strikingly poor pharmacokinetic properties after oral administration in their pure form, compared with herbal extracts (Table 1). For example, after oral administration of pure artemisinin, the exposure level of artemisinin in the bloodstream was more than 40-fold lower than that in the group treated with dried whole-plant *A. annua* [9].

Pharmacokinetic synergies among constituents in herbal extracts may occur during absorption, distribution, metabolism, and excretion. In fact, pharmacokinetic synergies during distribution and metabolism have been reported. For example, *Berberis* plants contain berberine, a substrate of p-glycoprotein (P-gp), and 5′-methoxyhydnocarpin, a strong P-gp inhibitor [18]. The use of 5′-methoxyhydnocarpin significantly increases the cellular uptake of berberine [18]. Moreover, artemisinin, a major active constituent of *A. annua* extracts, undergoes extensive metabolism mediated by cytochrome P450 enzymes (CYPs) (i.e., CYP2B6 and CYP3A4) [19]. A coexisting constituent in the extract, arteannuin B, inhibits hepatic CYP3A4 (IC_50_ 1.2 μM) and increases the AUC_0–t_ (2.1-fold) and peak concentration (C_max_, 1.9-fold) of oral artemisinin in mice [20]. However, some crucial factors in the synergistic interactions such as the solubility of the constituents, may affect absorption. In addition, these constituents showed the highest concentration in the gastrointestinal tract; hence, they are the most likely to have pharmacokinetic interactions there. For example, intestinal absorption plays a crucial role in differentiating between the pharmacokinetics of pure berberine and berberine in the *Coptidis Rhizoma* extract [21]. Therefore, this review mainly discusses the pharmacokinetic interactions during the absorption process.

Here, based on Web of Science searches using keywords such as “herb”, “herbal medicine”, “herbal extract”, “traditional Chinese medicine”, “intestinal absorption”, “pharmacokinetics”, “synergy”, “nanoparticle”, and “natural deep eutectic solvent”, specific searches were performed using the particular words and phrases related to the review. Studies published between 1995 and 2019 that examined pharmacokinetic synergies among constituents in herbal extracts during intestinal absorption were reviewed, with an emphasis on the formation of naturally occurring nanoparticles in herbal extracts and their roles in promoting absorption. Finally, we propose that herbal extracts are mixtures of bioactive compounds and pharmacokinetic synergists. In other words, there are natural, high-efficiency drug delivery systems in herbal extracts. This review aimed to broaden the general understanding of the pharmaceutical nature of herbal medicines and provide ideas and strategies for their further research and development.

## 2. Synergy Mechanisms in the Absorption Process

According to the Fick’s first law of diffusion, the absorption of a drug is directly proportional to its concentration in the gastrointestinal lumen (inclusive of dissolution, solubility, and stability of drug within the gastrointestinal tract) and permeability coefficient (inclusive of drug efflux) [22]. It was assumed that polyphenols and saponins are the key constituents in TCM remedies responsible for most of the observed biological effects [23]. Consistent with this hypothesis, the major marker compounds (>60%) for quality control among the 474 monographs of herbs usually used in the Chinese Pharmacopoeia are polyphenols, polysaccharides, and saponins [24]. However, these compounds are known for their poor solubility, permeability, and metabolic stability (i.e., they have significant oral bioavailability conundrum) [24]. How can compounds with poor pharmacokinetic properties act as the material basis for the efficacy of herbal medicines? Hence, it is reasonable to assume that coexisting constituents in herbal extracts may affect the intestinal absorption and, ultimately, the pharmacokinetics of these bioactive constituents through various mechanisms. The structures of some of the compounds discussed in this article are shown in Figure 1.

### 2.1. Synergies in Improving Water Solubility

Water solubility is a key determinant of oral drug bioavailability and has presented a major challenge in new drug research and development [25]. In 1996, Keung et al. [26] reported that daidzin, administered as an herbal extract, yielded an AUC value 10 times larger than that obtained with the same dose of pure daidzin. The researchers assumed that the enhanced solubility of daidzin in the extract accounted for the high bioavailability of daidzin [26]. Later studies revealed that other herb constituents such as hypericin [27] and berberine [12] have a much higher water solubility in the corresponding herbal extract.

Some small-molecule constituents in herbal extracts increase the solubility of coexisting bioactive constituents. For example, hyperoside (hyperin, quercetin 3-*O*-beta-d-galactoside) in St. John’s wort (*Hypericum perforatum* L.) increases the water solubility of hypericin by 400-fold [27]. Moreover, glycyrrhizic acid, a saponin produced by plants such as *Glycyrrhiza uralensis* Fisch [14], has a hydrophobic and hydrophilic component consisting of one triterpenoid aglycone molecule and two glycosyl groups, respectively. It can form “guest–host” complexes comprising one or two glycyrrhizic acid molecules per one guest molecule at a relatively low concentration (10^−5^–10^−3^ M) [28]. Given that the formation of complexes by glycyrrhizic acid increases simvastatin solubility by more than 100-fold [29], despite no currently available research report on this topic, it can be speculated that glycyrrhizic acid may also produce certain solubilizing effects with other TCM constituents. In addition, due to the hydrophobic interactions between triterpene moieties [30], glycyrrhizic acid forms micelles at critical concentrations above 10^−3^ M [28]. Micelles are rod-like and have an estimated radius and length of 1.5 nm and 21 nm, respectively [31]. Glycyrrhizic acid can thus act as a plant-derived surfactant [31].

It should be noted that the improved solubility of constituents in herbal extracts does not imply that these compounds are truly dissolved. In fact, only the apparent solubility of the compounds is increased. Herbal extracts are commonly used as dried powder that is reconstituted in water before oral administration. Given the high content of hydrophilic PMs in herbal extracts, a dry powder of herbal extracts may be considered as an amorphous solid dispersion of bioactive constituents [25,32]. Consequently, after dissolution in gastrointestinal fluids, supersaturated solutions of bioactive compounds may be formed [32]. In addition, some coexisting constituents such as plant-derived surfactants [33] or hydrophilic PMs may further act as precipitation inhibitors and prevent the formation of sediment, or more precisely, the crystallization of herbal extracts. However, direct evidence of this phenomenon is still lacking.

A novel natural deep eutectic solvent (NADES)-based mechanism has been proposed to explain synergies on water solubility in herbal extracts [2]. NADES was first recognized in 2011 as a botanical liquid medium that differs from water and lipids [34]. The solvents are supermolecules that induce hydrogen-bonding interactions between constituents in herbal extracts at certain ratios [35]. The constituents are basically plant-produced PMs including sugars, amino acids, choline, and organic acids [34]. In one study, a NADES increased the solubility of some lipophilic compounds by 18 to 460,000 times, compared with water [36]. The enhanced solubility may be beneficial in improving oral compound bioavailability. For example, a NADES composed of proline-glutamic acid (2:1) increased the solubility and doubled the bioavailability of rutin, compared with its water suspension [37]. Moreover, compared with its water suspension, a NADES composed of lactic acid:proline:malic:acid:water (1:0.2:0.3:0.5) increased the water solubility of oral berberine and increased its AUC value by four-fold [38].

However, given that dilution with water profoundly affects the dissolving capacity of NADES [35,36], the roles of NADES or its components in herbal aqueous extracts in improving solubility remain to be explored. The concentrations of NADES components in herbal aqueous extracts should be systematically and quantitatively analyzed, the ratios of the components should then be calculated, and the possibility of NADES formation and its effect on the solubility and bioavailability of active constituents could ultimately be determined.

### 2.2. Synergies in Inhibiting Intestinal Metabolism

Drug-metabolizing enzymes carry out the elimination of most drugs. CYPs, especially CYP3A4, and UDP-glucuronosyltransferases (UGTs), especially UGT1A1 and 2B1, are major phase I and phase II drug-metabolizing enzymes, respectively. The enzymes are majorly expressed in hepatocytes. However, intestinally expressed drug-metabolizing enzymes are also actively or even crucially involved in the pre-systemic metabolism of some oral drugs [39]. Therefore, intestinal absorption can be promoted or inhibited due to the interactions between co-administered constituents through the inhibition or induction of the enzymes. For example, flavonoids in the leaves of *A. annua* may increase the level of unchanged artemisinin that reaches the blood stream by suppressing CYPs [40]. Interestingly, the intestinal phase II metabolism of three constituents in the *Radix Scutellariae* extract, namely baicalein, wogonin, and oroxylin A, is inhibited by each constituent. For example, about 62.4% of pure baicalein was metabolized in the Caco-2 cell monolayer during transportation, but only 24.3% of baicalein in the mixture of baicalein, wogonin, and oroxylin A was metabolized [41]. Therefore, co-administration of these compounds significantly enhances their intestinal absorption by reducing intestinal metabolism [41]. Numerous drug-metabolizing enzyme inhibitors and inducers have been identified in herbal extracts, and have been found to affect the intestinal absorption and exposure levels of some oral drugs [42]. Therefore, it is reasonable to speculate that intestinal metabolism-based pharmacokinetic interactions between constituents in herbal extracts occur widely.

It should be noted that microbiota play important roles in the intestinal metabolism of some active constituents that belong to alkaloids, flavonoids, polyphenols, and terpenoids [24,43]. In particular, glycosides such as saponins, iridoid glycosides, and flavone glycosides are often metabolized to secondary glycosides and/or aglycones with better bioavailability by intestinal microbiota [44]. Surely, intestinal microbiota mediated metabolism is not limited to deglycosylation. For example, intestinal microbiota mediated transformations of phenylethanoid glycosides include degradation, reduction, hydroxylation, acetylation, hydration, methylation, and sulfate conjugation [45]. It is reasonable to assume that some coexisting constituents in herbal extracts can affect the intestinal metabolism, and subsequently the absorption of glycosides by regulating the intestinal microbiota. It was reported that ginseng polysaccharides promoted the intestinal biotransformation of ginsenosides Re and Rc via stimulating the growth of *Lactobacillus* spp. and *Bacteroides* spp., two strains of intestinal microbiota [44]. In addition, ginseng polysaccharides showed a prebiotic-like effect and enhanced the microbial deglycosylation and systemic exposure of Rb_1_ [46]. Furthermore, polysaccharides in the extract of *Ophiopogonis Radix* stimulated the gut microbiota-induced metabolism of ophiopogonins by increasing the activities of beta-d-glucosidase, beta-d-xylosidase, alpha-l-rhamnosidase, and beta-d-fucosidase [47].

Unfortunately, intestinal metabolism-mediated pharmacokinetic interactions among TCM constituents have not received enough attention, at least compared with hepatic metabolic pharmacokinetic interactions.

### 2.3. Synergies in Reducing Intestinal Efflux

Uptake drug transporters mediate both facilitated diffusion and active transport, thus aiding in the intestinal absorption of some oral drugs [48]. On the contrary, efflux transporters such as P-gp are abundantly expressed in enterocytes, preventing the cellular uptake of its substrates [48]. Furthermore, functional coupling may occur between intestinally expressed CYPs and P-gp [49]. For example, P-gp enhances intestinal drug metabolism by prolonging the access of drugs to CYP3A4 near the apical membrane of enterocytes and by decreasing their transport across cells [50].

The discovery of P-gp inhibitors from food and plant extracts has been going on for a long time. For example, it was reported in 1995 that grapefruit juice did not influence the pharmacokinetics of intravenous cyclosporine, but significantly increased the peak concentration (936 versus 1340 ng/mL) and area under the curve (6722 versus 10,730 ng h/mL of oral cyclosporine. In addition, grapefruit juice had no effect on the elimination half-life of oral cyclosporine [51]. The results showed that the improvement in the oral bioavailability of cyclosporine by grapefruit juice was related to the increase of cyclosporine absorption [51]. This study provides a valuable experimental design strategy for the discovery of P-gp inhibitors. Food derived compounds such as piperine [52], resveratrol [53], and capsaicin [54] were successively identified as P-gp inhibitors.

Many P-gp inhibitors have been identified in herbal extracts [42]. Moreover, many bioactive constituents in herbal extracts are P-gp substrates. Thus, coexisting constituents that are inhibitors of P-gp may increase the intestinal absorption of these bioactive constituents. For example, the exposure level of paclitaxel in rats receiving the extract of *Taxus yunnanensis* Cheng et L. K. Fu was more than seven times that in rats treated with pure paclitaxel [55]. The coexisting materials in *T. yunnanensis* extract significantly increases (by more than three times) the intestinal absorption of paclitaxel by inhibiting its intestinal efflux, which is majorly mediated by P-gp [55].

In addition, other efflux transporters such as breast cancer resistance protein (BCRP) and multidrug resistance protein 2 (MRP2) are also abundantly expressed in enterocytes, preventing the absorption of their substrates [48]. Similar to P-gp, some TCM constituents are the substrates of these efflux transporters [56,57], and others are their inhibitors [58,59,60]. Therefore, TCM constituents can interact with each other due to these transporters, thus affecting the intestinal absorption of the active constituents [61].

### 2.4. Synergies in Increasing Enterocyte Membrane Permeability

Most oral drugs are taken up by enterocytes through passive diffusion, suggesting that membrane permeability is a key factor determining the intestinal absorption of these drugs. Saponins are important bioactive constituents in herbal extracts [62]. They are amphiphilic molecules possessing a lipophilic aglycone and a hydrophilic sugar side chain [62]. Saponins have long been used to improve the transmembrane transport of drugs [63,64,65]. The effects of some saponins including glycyrrhizic acid [66,67,68], ginsenoside Rh2 [69], saikosaponin d [70], digitonin [71], and 3-*O*-β-d-glucopyranosyl platycodigenin [72] on membrane permeability are remarkable. For example, even at micromolar concentrations, glycyrrhizic acid increases the permeability (approximately 60%) and decreases the elasticity modulus (by an order of magnitude) of cell membranes [66]. Moreover, glycyrrhizic acid increases the maximum diffusion rate of formate ions through the cell membrane by 5.5 times [67]. Mechanisms including alteration of lipid mobility, formation of pores in the cell membrane, and decreased transport of water molecules into the cell membrane are known to be involved in the effects of glycyrrhizic acid [68].

### 2.5. Synergies in Opening Paracellular Tight Junctions Between Enterocytes

A tight junction (TJ) is composed of transmembrane proteins (such as occludin and claudins) and cytoplasmic plaque proteins (such as ZO-1, ZO-2, ZO-3, cingulin, and 7H6) [73]. In general, only low-molecular-weight compounds can pass through a TJ between enterocytes. Recently, however, chemical permeation enhancers (CPEs) have been used to transiently and reversibly open a TJ and prompt the intestinal absorption of oral drugs with low bioavailability [73]. CPEs may act directly and specifically on extracellular loops of TJ proteins and TJ-associated membrane microdomains [73]. Interestingly, some natural products including small-molecule constituents in herbal extracts, act as CPEs [74]. For example, Aloe vera gel and whole-leaf extract can promote drug-absorption [75] by modulating a tight junction [76]. Saponins can rapidly disrupt a TJ and subsequently increase the paracellular permeability of human intestinal Caco-2 cell monolayers [77]. In addition, sinomenine, an alkaloid extracted from the stem of *Sinomenium acutum* (Thunb.) Rehd. et Wils, acts as a CPE and promotes the absolute bioavailability of octreotide in rats and the transport rate of octreotide in Caco-2 cell monolayers [78,79]. In one study, homoharringtonine, a natural alkaloid produced by various *Cephalotaxus* species, increases intestinal epithelial permeability by modulating the protein expression and localization of claudin isoforms [80].

### 2.6. Synergies in Forming Naturally Occurring Nanoparticles

Artificially prepared nano-carriers have been used to deliver active constituents and fractions of herbal extracts due to their advantages in improving the intestinal absorption and pharmacokinetic properties of these constituents [81]. Interestingly, some naturally occurring nanoparticles with diverse sizes, shapes, and compositions have been identified in herbal extracts.

#### 2.6.1. Ubiquity of Naturally Occurring Nanoparticles in Herbal Extracts

In 1995, Groning et al. reported that particles with a mean size of approximately 200–300 nm were present in aqueous black tea extracts [82]. In 2003, nanoparticles with a mean diameter between 100 and 300 nm were found in a water solution of a dry extract of *H. perforatum* L., a medicinal plant [83]. Similarly, in 2008, Zhuang et al. reported that nanoscale “aggregates” (i.e., nanoparticles) were observed in 84 TCM extracts [84], indicating that naturally occurring nanoparticles are ubiquitous in herbal extracts.

#### 2.6.2. Isolation, Identification, and Composition of Natural Nanoparticles

Some natural nanoparticles have been isolated using a procedure that includes filtration, dialysis, and size exclusion chromatography [85,86]. The shape and size of natural nanoparticles can be characterized through atomic force microscopy (AFM), scanning electron microscopy (SEM), and dynamic light scattering/electrophoretic light scattering (DLS/ELS) [85,86]. The stability of natural nanoparticles can be estimated through Zeta potential analysis [86]. The chemical nature of natural nanoparticles can be determined using bicinchoninic acid (BCA) assay for proteins, anthrone-sulfuric acid assay for polysaccharides, and high-performance liquid chromatography or liquid chromatography tandem mass spectrometry (LC-MS/MS) for small-molecule compounds [85,86,87].

Most natural nanoparticles are amorphous [12], but some are spherical [83,88,89]. Similarly, they vary in size, but usually have a hydrodynamic radius (Rh) larger than 100 nm [84]. Although most natural nanoparticles have heterogeneous density [84,87], some have a density that is too low to allow their complete removal after super-speed centrifugation [84].

Some small-molecule constituents are involved in the formation of natural nanoparticles. For example, nanoparticles in aqueous black tea (fermented tea) extract are caffeine-polyphenol complexes [82]. Similarly, nanoscale aggregates in a water extract of *Pueraria lobata* (Willd.) Ohwi has been found to contain several small-molecule constituents [87]. In general, small molecules with structural and functional group motifs of polyphenols, polyionics, extended lipophiles, and extended conjugation are inclined to form aggregates at micromolar concentrations in water-rich conditions [90]. For example, phenolic compounds, especially flavonoids, have been found to form aggregates [91]. Eight bioactive small molecules (curcumin, kaempferol, physcion, silibinin, emodin, diphyllin, bufalin, and brazilin) isolated from herbal extracts form detectable colloidal aggregates [92]. Baicalin, derived from *Scutellariae Radix*, forms nanoparticles of approximately 100 nm with berberine hydrochloride derived from *Coptidis Rhizoma* [93]. Furthermore, mixtures of small molecules can be more prone to form aggregates [94]. Multiple forces (i.e., hydrophobic interaction, hydrogen bonds, electrostatic interactions, or Van der Waals attraction) are involved in this process [95]. In baicalin–berberine hydrochloride nanoparticle formation, electrostatic interaction drives the formation of one-dimensional complex units, and hydrophobic interaction induces further three-dimensional self-assembly [93].

However, as shown by genistein, aggregates formed by pure small molecules vary significantly in size and intensity, compared with aggregates in herbal extracts [87]. This indicates that other compounds in herbal extracts are involved in the formation of natural nanoparticles. In fact, some nanoparticles in herbal extracts are mainly composed of one or several plant-produced macromolecular metabolites including proteins [96,97,98], lipids [99,100], and polysaccharides [85,101].

Importantly, the macromolecular metabolites that form nanoparticles adsorb small molecules in herbal extracts such as those in the *Coptidis Rhizoma* extract [12] and *Ma-Xing-Shi-Gan-Tang* decoction [89]. In a *Ma-Xing-Shi-Gan-Tang* decoction extract, the majority of ephedrine (99.7%) and pseudoephedrine (95.5%) form nanoparticles rather than disperse freely in a water solution [89]. In addition, most shogaols in ginger extracts are not present in their free form, forming nanoparticles or microparticles instead [100]. Moreover, water-soluble compounds such as ephedrine attach to the surface of nanoparticles mainly via a secondary binding, whereas water-insoluble compounds such as aconitine are more likely to be integrated inside nanoparticles through binding with the hydrophobic domain of a protein [102].

Briefly, naturally occurring nanoparticles in herbal extracts could form aggregates of small-molecule constituents, macromolecular constituents, and macromolecular constituents adsorbed with small-molecule constituents [87].

#### 2.6.3. Factors Affecting Natural Nanoparticle Formation

The formation of natural nanoparticles is dependent on the chemical nature of a compound and several other factors, especially parameters of the extraction methods such as the type, temperature, pH values, ionic strength of extraction solvent, and the ratio of plant material to extraction solvent. For example, methanol is more effective than water in forming nanoparticles in a *H. perforatum* extract [83]. The mean diameters of natural nanoparticles decrease with increasing temperature and pH in aqueous infusions of the dried leaves of *Harungana madagascariensis* Lam. ex Poir. [88]. Moreover, the mineral composition of water (for example, Ca^2+^) affects the colloidal size and stability of a green tea [*Camellia sinensis* (L.) O. Ktze.] infusion [103]. The average size of nanoparticles formed in a water extract of *P. thomsonii* Benth (Fenge) [104] or *Rabdosia rubescens* (Hemsl.) Hara leaves [105] increases as the plant material concentration increases. Importantly, the mixing of several herbs allows easier nanoparticle formation [84]. In addition, a water solution of *H. perforatum* dried extract is more prone than its infusion to forming nanoparticles [83].

#### 2.6.4. Pharmacological Effects of Natural Nanoparticles

The bioactivities of natural nanoparticles are attributed to the bioactivities of the small molecule content. For example, nanoparticles in ginger extract exert hepatoprotective effects by activating the nuclear factor erythroid 2-related factor 2 (Nrf2) [100]. The bioactivities of these nanoparticles are majorly attributed to its shogaol content [100]. Moreover, the formed baicalin–berberine nanoparticles exhibited remarkable bacteriostatic activity [93]. However, PMs in natural nanoparticles still show certain intrinsic bioactivities. For example, nanoparticles isolated from green tea infusions contain polysaccharides, and hence exert an immuno-stimulatory effect by inducing the secretion of various cytokines [interleukin 6 (IL-6), tumor necrosis factor-α (TNF-α), granulocyte colony stimulating factor (G-CSF)], and chemokines [regulated upon activation normal T cell expressed and secreted factor (RANTES), IFN-γ-induced protein 10 (IP-10), and macrophage derived chemokine (MDC)] in RAW264.7 mouse macrophages [85]. This result provides a potential basis for the utilization of the multifunctional nanoparticles to improve antitumor efficacy in cancer immuno-chemotherapy [85].

Nanoparticles isolated from English Ivy showed a diameter of 108.8 ± 3.1 nm and a negative Zeta potential of 28.5 ± 3.2 mV [101]. The nanoparticles can be used for ultraviolet protection due to their optical properties and harmless properties [106]. Similarly, *Yunnan baiyao* is a TCM that has been used to treat wounds for over 100 years [107]. AFM images revealed that uniform nanofibers were present in relatively high abundance in a solution of this medicine [107]. The fibers were typically 25.1 nm in diameter and ranged from 86–726 nm in length after processing [93]. Due to the unique adhesive and structural properties of the nanofibers, these fibers can play a role in platelet aggregation, clotting formation, and wound healing [107].

#### 2.6.5. Pharmacokinetics of Natural Nanoparticles

Natural nanoparticles in herbal extracts are orally absorbable; moreover, they are formed and remain stable in simulated intestinal fluid [108]. It is well known that nanoparticles possess general bio-adhesion to biological mucosa including gastrointestinal mucosa [109]. Natural nanoparticles can pass through the monolayer of Caco-2 cells [84]. Moreover, ginger-derived nanoparticles are possibly internalized by Colon-26 cells through the phagocytosis pathway [110]. The size of natural nanoparticles is an important determinant of their intestinal absorption. For example, the absorption of *Pueraria thomsonii* Benth (Fenge) water decoction was improved with smaller nanoparticles [104].

Therefore, natural nanoparticles can be circulated through microvilli (M) cells in Peyer’s patches of mucosa-associated lymphoid tissue in the intestinal lumen via transcytosis [111]. Importantly, the lymphatic pathway indicates bypass hepatic first-pass elimination [112], which can help increase oral drug bioavailability. However, there are exceptions. The ginger-derived nanoparticles migrate into circulation through vascular vessels [100].

Natural nanoparticles may be stable in biological or even in whole animal milieus [113] including high-protein environments [114]. The results indicate that once the natural nanoparticles enter the circulation, they will keep their original size, that is, they will not be disrupted. In general, uptake by the reticular endothelial system (RES) such as macrophages will lead to the elimination of nanoparticles. However, as indicated in the study on nanoparticles formed in an aqueous extract of *R. rubescens* leaves, natural nanoparticles with a diameter of less than 100 nm are not easily eliminated by RES [105]. Finally, most of these natural nanoparticles may reach the drug target. For example, the ginger-derived nanoparticles were distributed mainly in liver tissues, where they protect against alcohol-induced liver damage by activating Nrf2 [100].

It should be noted that microparticles and precipitates are also formed in herbal extracts. A study showed that large aggregates, however, cannot pass through intact monolayers [84]. The formation of particles larger than 250 nm may have had a negative effect on the intestinal absorption of some compounds [113]. Nanoparticles with a diameter exceeding 200 nm are more likely to be swallowed by reticuloendothelial phagocytes (macrophages), and thus will be out of blood circulation [115]. For example, the formation of aggregates sized dozens of micrometers in Gegen (*Puerariae Lobatae Radix*) decoction led to a decreased plasma concentration of puerarin and daidzein, two major bioactive constituents in Gegen extract [116]. Ideally, the size of natural nanoparticles should be smaller than 50 nm [113]. Given that the size of natural nanoparticles is drug concentration-dependent [104,105], drug doses should be optimized for better intestinal drug absorption [116].

#### 2.6.6. Natural Nanoparticles in the Delivery of Active Constituents

Natural nanoparticles can act as drug carriers and contribute in the intestinal absorption of some small-molecule compounds in herbal extracts. For small-molecule compounds with low bioavailability, nanoparticles can help reduce first-pass elimination of the compounds and ultimately improve their bioavailability. For example, berberine has a bioavailability of as low as 0.36% [117], mainly due to its limited solubility [12], extensive first-pass metabolism [117], and efflux mediated by P-gp during intestinal absorption [118] and after hepatic distribution [119]. Proteinaceous nanoparticles in *Coptidis Rhizoma* absorb berberine and then promote its intestinal absorption [12]. The effects of the proteinaceous nanoparticles can explain, at least part, of the huge pharmacokinetic differences between pure berberine and berberine in the *Coptidis Rhizoma* extract [12]. Rb1 is a major active constituent in *Ginseng Radix et Rhizoma* extracts. Polysaccharides isolated from a *Ginseng Radix et Rhizoma* extract significantly facilitated Rb_1_ transport across the Caco-2 monolayer, increasing the *P*_app_ of Rb1 from 5.54 × 10^−7^ cm/s to 3.97 × 10^−6^ cm/s [46].

Nevertheless, in general, related researches are still insufficient. It is encouraging to find that the development of natural and efficient drug delivery systems using plant-derived nanoparticles has been proposed and practiced [120]. For example, nanoparticles isolated from green tea infusions [85,121], ivy [101], and ginger [110] have been successfully used to deliver doxorubicin, an antitumor drug.

## 3. Causes of Pharmacokinetic Synergies in Herbal Extracts: A Botanical Perspective

SMs are produced in response to stress such as ultraviolet radiation, drought, and temperature changes [122,123], and they play crucial roles in defense against pathogens [124,125]. For example, phenolic compounds are well-known antioxidants [126,127], whereas alkaloids [128] and saponins [129] have significant antimicrobial effects.

SMs can be synthesized or activated from inactive precursor compounds in response to attack by microorganisms [130]. Drug-metabolizing enzymes such as CYPs and UGTs are key players in this process [131]. SMs can also be synthesized prior to this attack. However, because most of these metabolites are harmful to the plant itself, after synthesis, they are transported to and restricted in specific sites such as vacuoles [132,133] for storage, and are released when the plant is under attack [134]. Drug transporters are closely involved in these processes [125,135]. From an evolutionary perspective, there are endless and fierce “arms races” between plants and pathogens [136]. For example, pathogens cause drug resistance by pumping out SMs [137], whereas plants produce novel SMs or efflux transporter inhibitors [138,139] to counter this resistance. Therefore, SMs are composed of not only the substrates of drug-metabolizing enzymes and transporter, but also their modulators.

In addition, in the case of drought stress, the formation of NADES due to the lack of water not only protects DNA and proteins [36], but also increases the solubility of insoluble flavonoids, thus enhancing their antioxidant ability [34].

In conclusion, herb extracts are organic and unified mixtures. Pharmacokinetic synergies between the constituents are the results of the physiological, pathological, and evolutional processes of the plant.

## 4. Discussion

As discussed above, herbal extracts can be considered as a mixture of bioactive compounds and pharmacokinetic synergists or natural pharmaceutical excipients. These pharmacokinetic synergists build natural, high-efficiency drug delivery systems in herbal extracts. As numerous major bioactive compounds in herbal extracts have been identified [5], it is time to systematically study natural pharmacokinetic synergists.

A synergy-directed fractionation strategy has been proposed and practiced in the search for pharmacodynamic synergists in herbal extracts [140]. To overcome inherent bias in the strategy (i.e., only limited compounds that were most easily isolated could be identified), a new biochemometrics strategy combining untargeted metabolomics with synergy-directed fractionation was recently developed [141]. The strategy was successfully used to identify pharmacodynamic synergists that enhanced the antimicrobial activity of berberine in *Hydrastis canadensis* (Goldenseal) [141]. Therefore, the strategy could also be used to identify pharmacokinetic synergists in herbal extracts.

What deserves looking forward is that bioactive constituents supplemented with plant-produced synergists would potentially produce novel herbal medicines with definite compounds, controllable quality, and remarkable pharmacological effects.

In summary, pharmacological losses of many herbal medicine-derived bioactive constituents are apparently attributed to losses of pharmacokinetic synergies after separation and purification. Coexisting plant-produced compounds including PMs and SMs affected the intestinal absorption, and ultimately, the pharmacokinetics of active constituents via various mechanisms (Figure 2) including by improving solubility, preventing drug-metabolizing enzyme- and efflux-drug-transporter-mediated first-pass elimination, increasing membrane permeability, opening paracellular tight junctions, and changing the forms and absorption of bioactive compounds (e.g., by forming naturally occurring nanoscale particles). These findings suggest that herbal extracts are, in fact, a mixture of bioactive compounds and pharmacokinetic synergists. This review contributed a broader understanding of the pharmaceutical nature of herbal medicines and provided ideas and strategies for their further research and development.

## Figures and Tables

**Figure 1 pharmaceutics-12-00128-f001:**
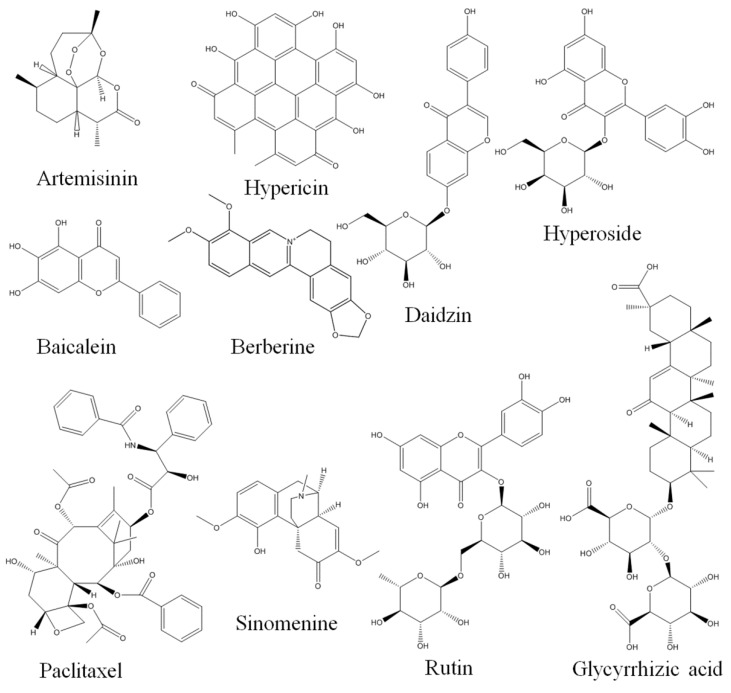
Structures of some compounds discussed in the manuscript.

**Figure 2 pharmaceutics-12-00128-f002:**
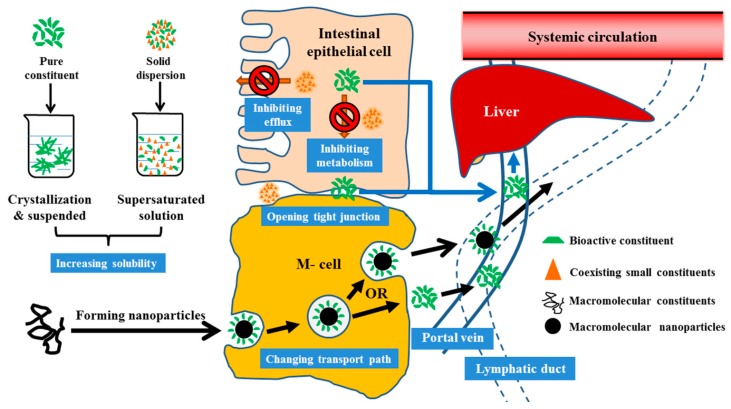
Potential pharmacokinetic synergies among constituents for increasing the intestinal absorption of active constituents in herbal extracts. Coexisting plant-produced compounds including primary and secondary metabolites affected the intestinal absorption, and ultimately, the pharmacokinetics of active constituents by improving solubility, inhibiting first-pass elimination mediated by drug-metabolizing enzymes and efflux-drug-transporter, increasing membrane permeability, opening paracellular tight junctions, and changing the forms and absorption of bioactive compounds (e.g., by forming naturally occurring nanoscale particles).

**Table 1 pharmaceutics-12-00128-t001:** Pharmacokinetic differences between some herbal extracts and their pure constituents.

Plants	TCM Names	Active Constituents	AUC_0–t extract_/AUC_0–t pure constituent_	References
*Aconitum carmichaelii* Debx.	Aconiti Lateralis Radix Praeparata	hypaconitine	2.7	[10]
*Artemisia annua* L.	Artemisiae Annuae Herba	artemisinin	>40	[9]
*Cnidium monnieri* (L.) Cuss.	Cnidii Fructus	osthole	>13.5	[11]
*Coptis chinensis* Franch.	Coptidis Rhizoma	berberine	15.3	[12]
*Gentiana manshurica* Kitag.	Gentianae Radix et Rhizoma	gentiopicroside	2.2	[13]
*Glycyrrhiza uralensis* Fisch.	Glycyrrhizae Radix et Rhizoma	liquiritigenin	133	[14]
		isoliquiritigenin	109	
*Panax ginseng* C. A. Mey.	Ginseng Radix et Rhizoma	ginsenoside Re	3.9	[15]
*Salvia miltiorrhiza* Bge.	Salviae Miltiorrhizae Radix et Rhizoma	cryptotanshinone	4.1	[16]
		tanshinone IIA	19.1	
*Schisandra chinensis* (Turcz.) Baill	Schisandrae Chinensis Fructus	schizandrin	2.2	[17]

AUC_0–t extract_ and AUC_0–t pure constituent_ indicate the exposure levels of active constituents in animals that received the oral herbal extracts or pure constituents, respectively. The ratios were calculated based on reported AUC (area under the curve) values or directly cited from the references.

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
