# Peer review of "Synergistic Mechanisms of Constituents in Herbal Extracts during Intestinal Absorption: Focus on Natural Occurring Nanoparticles"

_pharmaceutics, 2020, doi:10.3390/pharmaceutics12020128_

Round 1

Reviewer 1 Report

Title: “Synergistic effects of constituents in herbal extracts during intestinal absorption”

General comment.

The review was about pharmacokinetic synergies among constituents in herbal extract. In particular were analyzed different mechanisms that affect the bioactive intestinal absorption.

The argument is very interesting, but the authors could improve the work expanding their research and better rearrange the paragraph to clarify the focus of the review.

Minor comment.

Introduction: The concepts seems to be repetitive, also if were reported in different manner. For example, lines 74-75 “Pharmacological losses are closely associated with loss of pharmacokinetic synergies among constituents in herbal extracts” as concept that seems to repeat in all paragraph.

Please, rephrase it.

Major comment.

On the basis of which criteria were the plants chosen? Page 3 lines 99-102: it is clear that they are plants of the Chinese tradition. If so, the title should be changed. What time period the authors considered? The authors should expand the herbal extract list with other sites of research not reviewed on “Web of science” and amplify their research with other keywords such as “intestinal microbiota”. Paragraph 2: “Synergy mechanisms in the absorption process”. The title and the content were generalized. In this paragraph, the authors should better explain the problems of solubility, permeability, first passage, etc and link them to intestinal absorption. Also, if preferable added more plant examples and expand the comments. page 6: Much more space has been given to the topic on nanoparticles than to other topics. If you intend to maintain this structure, you need to change the title and highlighting it. Furthermore, it is necessary to better explain the effect of these nanoparticles intended as a synergistic effect of the components in herbal extract in intestinal absorption. Also these parts should be amplified with other herbal extracts.

Pag. 8. Paragraph 3. Please, amplify also these paragraph.

The English language will be revised.

Author Response

Dear reviewer,

Thank you for your valuable comments and suggestions on our manuscript.

Below are our responses to the comments in detail, which have been incorporated into the revised manuscript.

We hope that the revised manuscript is now suitable for publication.

Sincerely,

Bing-Liang Ma & Wei-Dong Zhang

Comment 1: Introduction: The concepts seems to be repetitive, also if were reported in different manner. For example, lines 74-75 “Pharmacological losses are closely associated with loss of pharmacokinetic synergies among constituents in herbal extracts” as concept that seems to repeat in all paragraph. Please, rephrase it.

Response 1: The constituents in herbal extracts may play synergistic effects in several aspects including pharmacy, pharmacokinetics and pharmacodynamics. This sentence attempts to explain the importance of pharmacokinetic synergy. We realized that two “loss” are used in the sentence, so following your suggestion, this sentence is modified as: “The decrease of pharmacological effects is closely related to the loss of pharmacokinetic synergies among constituents after the herbal extract is purified.”

Comment 2: On the basis of which criteria were the plants chosen? Page 3 lines 99-102: it is clear that they are plants of the Chinese tradition. If so, the title should be changed.

Response 2: This is a very valuable comment. There are no specific restrictions on the selection of plants in this review, as long as it meets the purpose of this review. It is undeniable that most of these plants are used as traditional Chinese medicine (TCM) in China, which may be related to the fact that synergy has been paid more attention in the field of traditional Chinese medicine. Considering that some non-TCM plants are included in this review, and the phenomena observed in TCM extracts are also applicable to other plants, we hope to keep the expression of “herbal extracts” in the manuscript.

Comment 3: What time period the authors considered?

Response 3: In the Introduction (paragraph 2, page 6), the following expression were added: “Studies published between 1995 and 2019 that examined pharmacokinetic synergies among constituents in herbal extracts during intestinal absorption were reviewed, with an emphasis on the formation of natural occurring nanoparticles in herbal extracts and their roles in promoting absorption.”

Comment 4: The authors should expand the herbal extract list with other sites of research not reviewed on “Web of science” and amplify their research with other keywords such as “intestinal microbiota”.

Response 4: In order to ensure the quality of researches, we only selectively review those studies published in journals included in Web of Science. It is possible that some high-level studies were not included in this review, which is a deficiency of this review. But we believe that the studies selected in this review should already be fairly representative.

Thanks for the suggestion for including studies on the “intestinal microbiota”. Comments on intestinal microbiota were added to the section “2.2. Synergies in inhibiting intestinal metabolism” as follows. “It should be noted that microbiota play important roles in the intestinal metabolism of some active constituents that belong to alkaloids, flavonoids, polyphenols, and terpenoids [24, 43]. Particularly, glycosides, such as saponins, iridoid glycosides and flavone glycosides, are often metabolized to secondary glycosides and/or aglycones with better bioavailability by intestinal microbiota [44]. Surely, intestinal microbiota mediated metabolism is not limited to deglycosylation. For example, intestinal microbiota mediated transformations of phenylethanoid glycosides include degradation, reduction, hydroxylation, acetylation, hydration, methylation, and sulfate conjugation [45]. It is reasonable to assume that some coexisting constituents in herbal extracts can affect the intestinal metabolism and subsequently the absorption of glycosides by regulating the intestinal microbiota. It was reported that ginseng polysaccharides promoted the intestinal biotransformation of ginsenosides Re and Rc via stimulating the growth of Lactobacillus spp. and Bacteroides spp., two strains of intestinal microbiota [44]. In addition, ginseng polysaccharides showed a prebiotic-like effect and enhanced the microbial deglycosylation and systemic exposure of Rb1 [46]. Furthermore, polysaccharides in the extract of Ophiopogonis Radix stimulated the gut microbiota-induced metabolism of ophiopogonins by increasing the activities of beta-D-glucosidase, beta-D-xylosidase, alpha-L-rhamnosidase and beta-D-fucosidase [47].”

Comment 5: Paragraph 2: “Synergy mechanisms in the absorption process”. The title and the content were generalized. In this paragraph, the authors should better explain the problems of solubility, permeability, first passage, etc and link them to intestinal absorption. Also, if preferable added more plant examples and expand the comments.

Response 5: This paragraph is amended as follows. “According to the Fick's first law of diffusion, the absorption of a drug is directly proportional to its concentration in the gastrointestinal lumen (inclusive of dissolution, solubility, and stability of drug within the gastrointestinal tract) and permeability coefficient (inclusive of drug efflux) [22]. It was assumed that polyphenols and saponins are the key constituents in TCM remedies responsible for most of the observed biological effects [23]. Consistent with this hypothesis, the major marker compounds (>60%) for quality control among the 474 monographs of herbs usually used in the Chinese Pharmacopoeia are polyphenols, polysaccharides, and saponins [24]. But these compounds are known for their poor solubility, permeability, and metabolic stability, i.e., they have significant oral bioavailability conundrum [24]. How can the compounds with poor pharmacokinetic properties act as the material basis for the efficacy of herbal medicines? Hence, it is reasonable to assume that coexisting constituents in herbal extracts may affect the intestinal absorption and, ultimately, pharmacokinetics of these bioactive constituents through various mechanisms.”

Comment 6: page 6: Much more space has been given to the topic on nanoparticles than to other topics. If you intend to maintain this structure, you need to change the title and highlighting it.

Response 6: The title was modified as: “Synergistic mechanisms of constituents in herbal extracts during intestinal absorption: Focus on natural occurring nanoparticles”.

Comment 7: Furthermore, it is necessary to better explain the effect of these nanoparticles intended as a synergistic effect of the components in herbal extract in intestinal absorption. Also these parts should be amplified with other herbal extracts.

Response 7: Unfortunately, very little research has been done on their synergistic effect on the absorption of small molecules in herbal extracts. We look forward to further related researches. The following discussion was added: “But in general, related researches are still insufficient. By the way, it's encouraging to find that the development of natural and efficient drug delivery systems using plant-derived nanoparticles has been proposed and practiced [121]. For example, nanoparticles isolated from green tea infusions [85, 122], ivy [101], and ginger [110] have been successfully used to deliver doxorubicin, an antitumor drug.”

Comment 8: Pag. 8. Paragraph 3. Please, amplify also these paragraph.

Response 8: The paragraph was modified as follows: “Natural nanoparticles may be stable in biological or even in whole animal milieus [113], including high-protein environments [114]. The results indicate that once the natural nanoparticles enter the circulation, they will keep their original size, that is, they will not be disrupted. In general, uptake by the reticular endothelial system (RES) such as macrophages will lead to the elimination of nanoparticles. However, as indicated in the study on nanoparticles formed in an aqueous extract of R. rubescens leaves, natural nanoparticles with a diameter of less than 100 nm are not easily eliminated by RES [105]. Finally, most of these natural nanoparticles may reach the drug target. For example, the ginger-derived nanoparticles were distributed mainly in liver tissues, where they protect against alcohol-induced liver damage by activating Nrf2 [100].”

Comment 9: The English language will be revised.

Response 9: This thesis has been polished carefully.

Reviewer 2 Report

For section 2, the quantitative value for the synergistic effects of constituents in herbal extracts should be presented. Should be presented the chemical structures for constituents in herbal extracts discussed. Should be clarified the synergistic effects of constituents in herbal extracts using Table. Should be explained in molecular level for the synergistic effects of constituents in herbal extracts. Why solid dispersion presented in Fig 1? How is it related to synergistic effects of constituents in herbal extracts? What is forming nanoparticles? Why bioactive constituent (green) attached in nanoparticles? Is the bioactive constituent distributed only in the nanoparticles?

Author Response

Dear reviewer,

Thank you for your valuable comments and suggestions on our manuscript.

Below are our responses to the comments in detail, which have been incorporated into the revised manuscript.

We hope that the revised manuscript is now suitable for publication.

Sincerely,

Bing-Liang Ma & Wei-Dong Zhang

Comment 1: For section 2, the quantitative value for the synergistic effects of constituents in herbal extracts should be presented.

Response 1: We agree with you that it is necessary to evaluate the synergy quantitatively. We have provided some quantitative information in the submitted paper. For example, "hyperside, …increases the water solubility of hypericin by 400-fold", “a NADES increased the solubility of some lipopic complexes by 18 to 460000 times”, “a NADES…increases the solubility and doubled the bioavailability of rutin”, “a NADES…increases the water solubility of oral berberine and increases its AUC value by four-fold", “the coexisting materials in T. yunnanensis extract significantly increases (by more than three times) the intestinal absorption of paclitaxel”, “glycyrrhizic acid increases the maximum diffusion rate of formate ions through the cell membrane by 5.5 times”, “Polysaccharides…increasing the Papp of Rb1 from 5.54×10−7 cm/s to 3.97×10−6 cm/s”.

In the revised manuscript, other quantitative information was provided. For example, “about 62.4% of pure baicalein was metabolized in Caco-2 cell monolayer during transportation, but only 24.3% of baicalein in the mixture of baicalein, wogonin, and oroxylin A was metabolized [41].” For another example, “grapefruit juice…significantly increased the peak concentration (936 versus 1340 ng/ml) and area under the curve (6722 versus 10730 ngï¹’h/ml) of oral cyclosporine.”

Comment 2: Should be presented the chemical structures for constituents in herbal extracts discussed.

Response 2: Thank you for this suggestion. The structure of some compounds discussed in this article is provided in figure 1 in the revised manuscript.

Comment 3: Should be clarified the synergistic effects of constituents in herbal extracts using Table.

Response 3: Tables have an intuitive advantage, but to facilitate discussion, we hope to retain the existing architecture.

Comment 4: Should be explained in molecular level for the synergistic effects of constituents in herbal extracts.

Response 4: This is a very good suggestion. For example, it was reported that in baicalin-berberine hydrochloride nanoparticle formation, electrostatic interaction drives the formation of one-dimensional complex units, and hydrophobic interaction induces further three-dimensional self-assembly. We have cited the results and other mechanisms studies in this review. But it is unfortunate to find that there are very few reports on the mechanism research at the molecular level. This should be an area worthy of further study in the future.

Comment 5: Why solid dispersion presented in Fig 1? How is it related to synergistic effects of constituents in herbal extracts?

Response 5: We have discussed this issue in section “2.1. Synergies in improving water solubility (page 5, paragraph 2)”. In brief, the dry powder of herbal extracts may be considered an amorphous solid dispersion of bioactive constituents. But as discussed in the manuscript, direct evidence is still lacking.

Comment 6: What is forming nanoparticles? Why bioactive constituent (green) attached in nanoparticles? Is the bioactive constituent distributed only in the nanoparticles?

Response 6: We have discussed these issues in section “2.6. Synergies in forming naturally occurring nanoparticles”. In brief, â‘  (page 8, paragraph 2&3) some small-molecule constituents are involved in the formation of natural nanoparticles, but some nanoparticles in herbal extracts are mainly composed of one or several plant-produced macromolecular metabolites, including proteins, lipids, and polysaccharides; â‘¡ (page 8, paragraph 2) multiple forces, i.e. hydrophobic interaction, hydrogen bonds, electrostatic interactions, or Van der Waals attraction, are involved in the formation of nanoparticles; â‘¢ (page 8, paragraph 4) surely, only part of the bioactive constituents are distributed in the nanoparticles in most cases. But “in a Ma-Xing-Shi-Gan-Tang decoction extract, the majority of ephedrine (99.7%) and pseudoephedrine (95.5%) form nanoparticles rather than disperse freely in a water solution [89]. In addition, most shogaols in ginger extracts are not present in their free form, forming nanoparticles or microparticles instead [100].”

Reviewer 3 Report

The review is well written and give the essential information on effects of herbal extracts on intestinal drug absorption.  The following are recommendations to improve the paper:

1)  Page 5, section 2.3:  please give a short description of how plant based interactions with efflux transporters were discovered by the influence of grapefruit juice on cyclosporin absorption in a clinical trial.  Also other prominent phytochemicals can be given as examples in this section such as capsaicin (Bedada et al., 2017, Drug Dev Ind Pharm), resveratrol (Jia et al., 2016, Toxicol. Appl. Pharmacol) and piperine (Jin et al., 2010, J. Food Sci).

2) Page 5, section 2.5:  please add more prominent examples of plant extracts that mediated tight junction opening for paracellular transport such as Aloe vera gel and whole leaf extract (Haabroek et al., 2019, Pharmaceutics).

3)  Page 2, line 64: please clarify the sentence to ensure it states that the complexity of plant phytochemistry impacts on the composition, but quality of plant medicines can still be ensured by measurement of marker molecules etc.

4)  Page 4, line 172:  "the absorption" could rather be described as "the unchanged drug that reaches the blood stream".

Author Response

Dear reviewer,

Thank you for your valuable comments and suggestions on our manuscript.

Below are our responses to the comments in detail, which have been incorporated into the revised manuscript.

We hope that the revised manuscript is now suitable for publication.

Sincerely,

Bing-Liang Ma & Wei-Dong Zhang

Comment 1: Page 5, section 2.3:  please give a short description of how plant based interactions with efflux transporters were discovered by the influence of grapefruit juice on cyclosporin absorption in a clinical trial. Also other prominent phytochemicals can be given as examples in this section such as capsaicin (Bedada et al., 2017, Drug Dev Ind Pharm), resveratrol (Jia et al., 2016, Toxicol. Appl. Pharmacol) and piperine (Jin et al., 2010, J. Food Sci).

Response 1: These works are cited in the revised paper as follows. “The discovery of P-gp inhibitors from food and plant extracts has been going on for a long time. For example, it was reported in 1995 that grapefruit juice did not influence the pharmacokinetics of intravenous cyclosporine, but significantly increased the peak concentration (936 versus 1340 ng/ml) and area under the curve (6722 versus 10730 ngï¹’h/ml) of oral cyclosporine. In addition, grapefruit juice had no effect on the elimination half-life of oral cyclosporine [51]. The results showed that the improvement of oral bioavailability of cyclosporine by grapefruit juice was related to the increase of cyclosporine absorption [51]. This study provides a valuable experimental design strategy for the discovery of P-gp inhibitors. Food derived compounds such as piperine [52], resveratrol [53], and capsaicin [54] were successively identified to be P-gp inhibitors.”

Comment 2: Page 5, section 2.5:  please add more prominent examples of plant extracts that mediated tight junction opening for paracellular transport such as Aloe vera gel and whole leaf extract (Haabroek et al., 2019, Pharmaceutics).

Response 2: The studies of Haabroek were cited as “For example, Aloe vera gel and whole-leaf extract can promote drug-absorption [75] by modulating tight junction [76].”

Comment 3: Page 2, line 64: please clarify the sentence to ensure it states that the complexity of plant phytochemistry impacts on the composition, but quality of plant medicines can still be ensured by measurement of marker molecules etc.

Response 3: The sentence was modified as follow: “Owing to their intrinsic complexity, the quality control of herbal medicines is very challenging”.

Comment 4: Page 4, line 172:  "the absorption" could rather be described as "the unchanged drug that reaches the blood stream".

Response 4: The sentence was modified as “For example, flavonoids in the leaves of A. annua may increase the level of unchanged artemisinin that reaches the blood stream by suppressing CYPs [40]”.

Reviewer 4 Report

This is an interesting review and it is generally well written.

However, in some parts it resembles more a list of activities than a critical review. The review will need to have greater depth and a more critical appraisal in terms of what these data actually mean in scientific terms as well as on their clinical relevance.

It would benefit from a more systematic search of the relevant literature. See for example:

Houghton P., and Pulok M, Synergy and polyvalence: paradigms to explain the activity of herbal products, in “Evaluation of Herbal Medicinal Products”, eBook, ISBN 9780853699217, Published Aug 2009).

Rasoanaivo et al., Malaria Journal 2011, 10(Suppl 1):S4, 1-12, Whole plant extracts versus single compounds for the treatment of malaria: synergy and positive interactions

Waldmann et al., Mol. Pharmaceutics 2012, 9, 815−822, Provisional Biopharmaceutical Classification of Some Common Herbs Used in Western Medicine;

Yang et al., Fitoterapia 92 (2014) 133–147. Synergy effects of herb extracts: Pharmacokinetics and pharmacodynamic basis

Zhou X. et al., Frontiers in Pharmacology, 2016, doi: 10.3389/fphar.2016.00201, Synergistic Effects of Chinese Herbal Medicine: A Comprehensive Review of Methodology and Current Research

Yuan et al., Molecules 2017, 22, 1135, How Can Synergism of Traditional Medicines Benefit from Network Pharmacology

Caesar L and Cech N. Nat. Prod. Rep., 2019, 36, 869–888. Synergy and antagonism in natural product extracts: when 1 + 1 does not equal 2.

Sun et al., Evidence-Based Complementary and Alternative Medicine, 2019, Article ID 1983780, 16 pages, https://doi.org/10.1155/2019/1983780. Influence Factors of the Pharmacokinetics of Herbal Resourced Compounds in Clinical Practice

In addition, since more than half of the manuscript describes synergies and factors affecting formation and effectiveness of naturally occurring nanoparticles, this should be also reflected on the title, the abstract and the introduction of the manuscript

Author Response

Dear reviewer,

Thank you for your valuable comments and suggestions on our manuscript.

Below are our responses to the comments in detail, which have been incorporated into the revised manuscript.

We hope that the revised manuscript is now suitable for publication.

Sincerely,

Bing-Liang Ma & Wei-Dong Zhang

Comment 1: This is an interesting review and it is generally well written. However, in some parts it resembles more a list of activities than a critical review. The review will need to have greater depth and a more critical appraisal in terms of what these data actually mean in scientific terms as well as on their clinical relevance.

It would benefit from a more systematic search of the relevant literature. See for example:

Houghton P., and Pulok M, Synergy and polyvalence: paradigms to explain the activity of herbal products, in “Evaluation of Herbal Medicinal Products”, eBook, ISBN 9780853699217, Published Aug 2009).

Rasoanaivo et al., Malaria Journal 2011, 10(Suppl 1):S4, 1-12, Whole plant extracts versus single compounds for the treatment of malaria: synergy and positive interactions

Waldmann et al., Mol. Pharmaceutics 2012, 9, 815−822, Provisional Biopharmaceutical Classification of Some Common Herbs Used in Western Medicine;

Yang et al., Fitoterapia 92 (2014) 133–147. Synergy effects of herb extracts: Pharmacokinetics and pharmacodynamic basis

Zhou X. et al., Frontiers in Pharmacology, 2016, doi: 10.3389/fphar.2016.00201, Synergistic Effects of Chinese Herbal Medicine: A Comprehensive Review of Methodology and Current Research

Yuan et al., Molecules 2017, 22, 1135, How Can Synergism of Traditional Medicines Benefit from Network Pharmacology

Caesar L and Cech N. Nat. Prod. Rep., 2019, 36, 869–888. Synergy and antagonism in natural product extracts: when 1 + 1 does not equal 2.

Sun et al., Evidence-Based Complementary and Alternative Medicine, 2019, Article ID 1983780, 16 pages, https://doi.org/10.1155/2019/1983780. Influence Factors of the Pharmacokinetics of Herbal Resourced Compounds in Clinical Practice.

Response 1: We have carefully read the papers you recommended. We made great efforts to modify our submitted manuscript. Particularly, we enriched the background introduction and strengthened the analysis of the literatures cited. Please refer to the revised manuscript.

Comment 2: In addition, since more than half of the manuscript describes synergies and factors affecting formation and effectiveness of naturally occurring nanoparticles, this should be also reflected on the title, the abstract and the introduction of the manuscript.

Response 2: The title was modified as: “Synergistic mechanisms of constituents in herbal extracts during intestinal absorption: Focus on natural occurring nanoparticles”.  The sentence “This review will focus on explaining this new synergistic mechanism” was added to the Abstract. The sentence “Studies published between 1995 and 2019 that examined pharmacokinetic synergies among constituents in herbal extracts during intestinal absorption were reviewed, with an emphasis on the formation of natural occurring nanoparticles in herbal extracts and their roles in promoting absorption” was added to the Introduction section.

Round 2

Reviewer 1 Report

The authors answered the questions. They also expanded and arranged the work as required.

Reviewer 2 Report

I agree the publication of this manuscript in Pharmaceutics.

Reviewer 4 Report

The manuscript has been substantially improved after being revised according to reviewers' comments

Moderate english changes may be needeed e.g. text in lines 406-410 "But in general, related researches are still insufficient. By the way, it's encouraging to find that the development of natural and efficient drug delivery systems using plant-derived nanoparticles has  been proposed and practiced [121]. For example, nanoparticles isolated from green tea infusions [85, 408 122], ivy [101], and ginger [110] have been successfully used to deliver doxorubicin, an antitumor drug"

should be better written as

"In general, related research studies remain insufficient. However, it is encouraging that the development of efficient drug delivery systems using plant-derived nanoparticles has been not only proposed, but alo used in practice [121]. For example, nanoparticles isolated from green tea infusions [85, 408 122], ivy [101], and ginger [110] have been used successfully to deliver the anti-cancer drug, doxorubicin.